# Characterization of two linear epitopes SARS CoV-2 spike protein formulated in tandem repeat

Simson Tarigan[1]*, N. L. P. Indi Dharmayanti[1], Dianita Sugiartanti[2], Ryandini Putri[1], Andriani[1], Harimurti Nuradji[1], Marthino Robinson[3], Niniek Wiendayanthi[3], Fadjry Djufri[4]

**1** Research Organization for Health, National Research and Innovation Agency (BRIN), Jakarta, Indonesia, **2** Indonesian Research Institute for Veterinary Sciences, Bogor, Indonesia, **3** Bogor Municipality Public Hospital (RSUD Kota Bogor), Bogor, Indonesia, **4** Indonesian Agency for Agricultural Research and Development, Jakarta, Indonesia

* sitariganta@gmail.com

**Data Availability Statement:** All relevant data are within the paper.

**Funding:** This work was supported by the DIPA Badan Litbang Pertanian, Kantor Pusat Jakarta No.

## Abstract

The vital roles of diagnostic tools and vaccines are prominent in controlling COVID-19. Spike protein of the SARS CoV-2, specifically the epitopes in that protein, are the critical components of the vaccines and immunological diagnostic tools. Two epitopes in the spike protein, the S14P5 and S21P2, identified previously are of great interest because they are linear and elicit neutralizing antibodies. The present study formulated each epitope in the tandem-repeat structure to increase their immunogenicity and facilitate their production. The tandem repeats (TR) were expressed efficiently in *E. coli*, yielding 58 mg and 46 mg per liter culture for TR-S14P5 and TR-S212, respectively. ELISA using either one of the repeating epitopes can be used as a serological test to identify individuals infected by the SARS-CoV-2 virus. The area under curves (AUC), based on testing 157 serum samples from COVID-19 patients and 26 from COVID-19-free individuals, were 0.806 and 0.889 for TR-S14P5 and TR-S21P2-based ELISAs, respectively. For 100% diagnostic specificity, the sensitivity was only 70%. The low sensitivity supposedly resulted from some samples being from early infection prior to antibody conversion. Both recombinant epitopes were highly immunogenic in rabbits, and the immune sera recognized inactivated SARS CoV-2 virus in dot-blot assays. These antibodies should be useful as a reagent for detecting SARS-CoV-2 antigens. Furthermore, the TR-S14P5 and TR-S21P2, being conserved and denaturation-resistant, are envisaged to be ideal for intra-nasal vaccines, which are required to complement current COVID-19 to overcome rapidly mutated SARS CoV-2.

## Introduction

The worldwide suffering caused by COVID-19 has been unprecedented. Within 30 months since first identified in December 2019, the SARS-CoV-2 virus has infected at least 516,476,402 people globally, including 6,258,023, resulting in death (https://COVID19.who.int, 12 May 2022). The high morbidity and mortality rates are because the disease was

018.09.1.411971/2021, Kode MAK: 1809. AEA.501.051.A.522191 The funders had no role in study design, data collection and analysis, decision to publish, or preparation of the manuscrip.

**Competing interests:** The authors have declared that no competing interests exist.

previously unknown, thereby no efficacious treatment and preventive measures, and the disease spreads rapidly.

SARS CoV-2 virus, the cause of COVID-19, a member of the betacoronavirus genus, comprises 26–32 kb positive single-stranded RNA, four major structural proteins (spike, nucleocapsid, membrane, and envelope proteins) and 16 non-structural proteins. The spike protein has received the most attention because this protein plays an essential role in initiating infection, pathogenicity, transmission, and evolution. The spike, a 1273-amino-acid-long glycoprotein, is cleaved by proteases at residue 685/686 into sub-unit $S_1$ and $S_2$. The two sub-units, however, remain noncovalently bound in pre-fusion metastable conformation. The $S_1$ contains the signal peptide, N-terminal domain (NTD) and receptor-binding domain (RBD). The NTD determines the host range, whereas the RBD recognizes and binds to the human-angiotensin-converting enzyme-2 (hACE2) receptor. The $S_2$ subunit, which functions in host-viral membrane fusion and cell entry, comprises fusion peptide (FP), heptad repeat (HR)1, HR2, transmembrane (TM) and cytoplasmic tail (CT). All parts of $S_2$ involve in the host-viral membrane fusion. In addition to membrane fusion, TM and CT play vital roles in spike protein trimerization and anchoring the trimeric spike proteins [1, 2].

Throughout 2021 the World Health Organization (WHO) granted ten COVID-19 vaccines for emergency use. The vaccines include two mRNA platforms (Pfizer BioNTech Comirnaty and Moderna Spikevax), three adeno-vector platforms (Oxford AstraZeneca Vaxzevria, Covishield Oxford AstraZeneca, and Janssen Johnson & Johnson Jcovden), three inactive viruses (Sinopharm Covilo, Sinovac CoronaVac, and Bharat BiotechCovaxin, and two protein-subunit platforms (Novavax Nuvaxovid and COVOVAX Novavax) (https://covid19.trackvaccines.org/agency/who/). Massive vaccination in many countries has led to a marked reduction in Covid-19 cases [3].

However, while the implementation of vaccinations was still in full swing, a new variant of concern (VOC) SARS CoV2 emerged and circulated rapidly in succession, Alpha, Beta, Gamma, Delta and Omicron variants. The vaccine's effectiveness against those variants decreased progressively, from a slight decrease against the Alpha variant to a practically ineffective against the Omicron variant [4, 5]. The time from the first report of SARS CoV-2 to the emergence of the Omicron variant was only about two years [6–9]. This means that mutations leading to immune evasion in SARS-CoV-2 occur rapidly.

The ineffectiveness of the vaccines was associated with the fact that the vaccines rely on spike protein which is highly prone to mutation [10]. Guruprasad recently analysed the amino acid sequence of 303,250 spike proteins deposited in the NCBI virus database on 2 November 2021 and found that 96.5% of the proteins had mutated to various degrees. The domains of the protein were not equally susceptible to mutation; the N terminal (residues 1–302), the receptor binding (333–327), and the $S_1^D$ (residues 594–674) domains were the most frequently to undergo mutations [11].

Maintaining the effectiveness of the vaccines required updating the vaccines following the mutation of the virus in the fields. However, updating vaccines too frequently is not feasible. An alternative for maintaining vaccine effectiveness is to generate vaccines from conserved epitopes of the virus [12].

Poh *et al*. identified two conserved, linear epitopes, S15P5 and S21P2, in the spike protein eliciting neutralising antibodies. Epitope S14P5 is located in the $S_1$ subunit downstream of RBD, whereas epitope S21P2 is in the $S_2$ subunit part of the fusion peptide [13]. The neutralising activity of the antibodies supposedly resulted from steric hindrance. The S14P5 epitope (residues 553–571) and S21P2 epitope (residues 808–827) were relatively conserved [11]. The molecularly conserved structure of the epitopes is in line with the findings that S14P5 and S21P2 peptides have been used widely to detect antibodies in humans resulting from infection

with unidentified variants of SARS CoV-2 in Shenzhen, China [14], Singapore [15, 16] or Indiana, USA [17], or vaccination with BNT162b2 mRNA COVID-19 vaccine (BioNTech-Pfizer) [18].

Besides being proven useful for diagnosis, the S14P5 and S21P2 epitopes may also have the potential to be developed into vaccines. To be used as vaccines, however, the epitopes require some manipulation to increase their capacity to induce immune responses and to ease their production. Without structural modification, the immune responses elicited by the epitopes may not adequately inhibit infection. In the present study, the S14P5 and S21P2 epitopes were formulated in tandem repeat and expressed in *E. coli* to increase their immunogenicity and ease of production. Besides increasing antigenicity and immunogenicity, tandem repeat structure also facilitates the use of the epitopes in immunoassays because it facilitates the immobilization of the epitopes on hard supports. The recombinant proteins should resist denaturation because they are composed of linear epitopes. Being denaturation resistant, vaccines generated from the recombinant repeat epitopes should not require a cumbersome and costly cold chain. In addition, the denaturation-resistant property of the recombinant repeat epitope supports their use for intranasal vaccines. It is expected that vaccines developed from the recombinant S14P5 or S21P2 tandem repeat could complement the current Covid-19 vaccines to overcome the rapidly mutated SARS-COV-2.

## Materials and methods

### Construction of recombinant expression plasmid

As indicated in the previous study, the amino acid sequence of epitope S14P5 is "TESNKKFLPFQQFGRDIAD", and that of S21P2 is "DPSKPSKRSFIEDLLFNKVT" [13]. The TR-S14P and TR-21P2 were constructed by four tandem repeat epitopes, S14P5 or S21P4, and a flexible peptide linker (GGGS) was inserted between the epitopes (Fig 1A). The encoding genes for those tandem repeats were synthesized and inserted into expression plasmid pET30a (+) (Genscript USA, Inc). The recombinant plasmids were transformed into a BL21 strain of *E. coli* which had been made competent by a previous method [19]. The transformed-competent cells were plated on LB agar containing 30 μg/ml of kanamycin. After overnight incubation, 15 colonies were randomly selected and grew overnight in LB broth containing kanamycin 30 μg/ml (LB broth-$_{KAN}$). Fresh LB broth- $_{KAN}$ was inoculated with the overnight culture (1: 100 dilution) and grew at 37˚C with 200 rpm shaking. At log-phase growth (OD$_{600}$ = 0.4), the culture was induced with 1 mM IPTG. The cells were pelleted and solubilized two hours after the induction in an SDS-PAGE sample buffer. Pairs of samples, with and without IPTG induction, were run side by side in the SDS PAGE, and proteins were stained with Coomassie blue. A colony from each S14P5 or S21P2-transformed *E. coli* that expressed the highest recombinant protein was selected for recombinant epitope production.

Fresh LB broth-$_{KAN}$ was inoculated with the selected colonies, incubated and induced with IPTG as previously. Recombinant proteins were purified with a Ni-NTA agarose column (Thermo Fisher Scientific). The bacterial pellet was first suspended in lysis buffer (6M guanidine-HCl, 500 mM NaCl, 20 mM Na-phosphate, pH 8), sonicated, and then clarified at 6 000 x *G* for 15 min. The supernatant was subjected to a Ni-NTA column, and unbound proteins were washed off successively with 8 M urea and 40 mM imidazole, 500 mM NaCl, 20 m*M* Na-phosphate, pH 7.8. Proteins specifically bound to the resin were eluted with 500 mM imidazole, 500 mM NaCl, 20 mM Na-phosphate, pH 8. The eluted protein was determined its concentration spectrophotometrically at 280 nm, checked its purity with SDS PAGE, and kept at -20˚C until used. Assessment of purity was based on the thickness of the Coomassie-blue-stained target protein band relative to the contaminant bands.

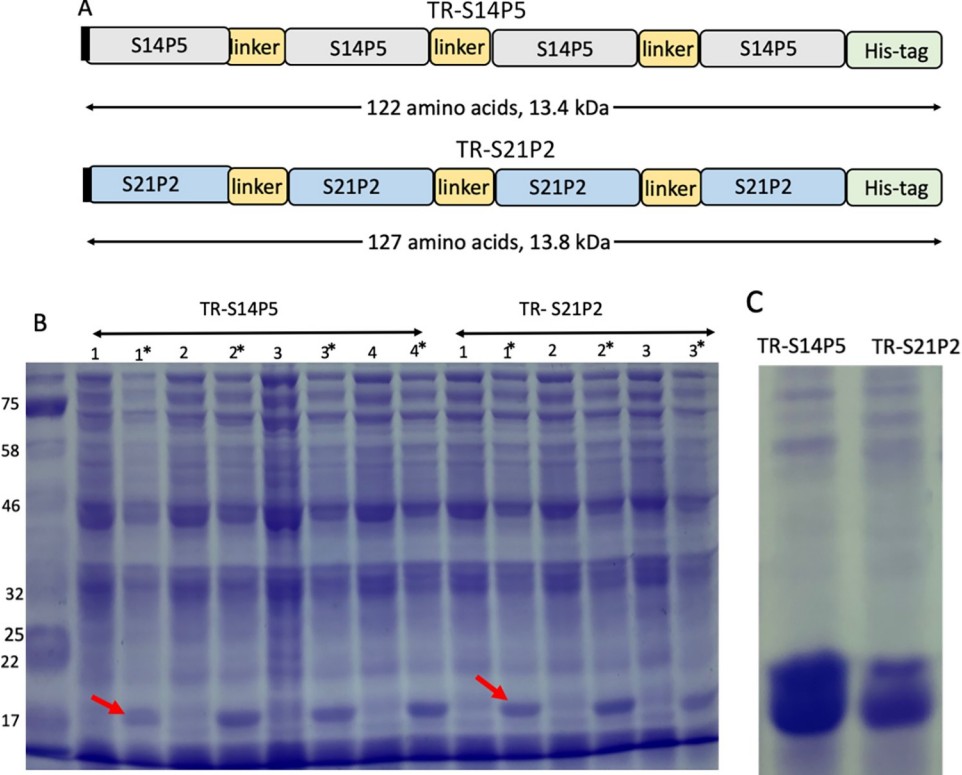

**Fig 1. Expression of repeat linear epitope S14P5 and S21P2 in *E. coli*.** (A): Schematic diagram of repeat linear epitope S14P5 and S21P2. (B): Protein profiles of transformed *E. coli* before and after (*) IPTG induction. Arrows are to indicate the repeat linear epitope. (C): Purification of Repeat epitopes from the lysate of transformed bacteria after IPTG induction.

## Serum samples and Enzyme-linked immunosorbent assay (ELISA)

Negative control serum, pooled sera from 10 persons free of COVID-19, and positive control serum from 8 persons recovered from clinical, PCR-confirmed COVID-19, prepared from our previous studies, were used in the current study. Serum samples collected from 158 COVID-19-confirmed patients being treated at the Bogor Municipal Hospital and 26 from COVID-19-free persons (PCR confirmed and no signs of any disease at least two weeks previously) were allocated for this study. The protocol of collection and the use of the human sera were approved by the Ethical Committee of the Bogor Municipal Hospital (SK No: 002/Sekt-Kepres/x/2020). Written informed consent was obtained from participants in accordance with the tenets of the Declaration of Helsinki.

## ELISA

The ELISA was carried out according to a previous protocol with some modifications [20]. The recombinant protein, TR-S14P5 or TR-S21P, was dissolved at 20 μg/ml in 0.1 M carbonate buffer, pH 9.6. A volume of 100 μl of the protein suspension was added to each well of a maxi-sorb Nunc plate (Sigma) and then incubated at 4˚C overnight. After the wells were blocked with 0.1% BSA, 100 μl serum samples diluted 1:100 in ELISA buffer (0.7 M NaCl, 0.05 M EDTA, 3% Triton X-100, 3% Tween-20, 2% non-fat skim milk, 5% goat serum, 0.1 M Tris, pH 7.4) was added. Positive and negative control sera diluted similarly were added respectively to four wells. After 2 hours of incubation, the wells were washed four times with a stringent

washing buffer (640 μM NaCl, 3 μM KCl, 8 μM $Na_2HPO_4$, 1.5 μM $KH_2PO_4$ and 5% Tween-20). A solution of anti-human-IgG-horseradish-peroxidase conjugate (Sigma Chemicals, St Louis, MO, USA) diluted in ELISA buffer was added and incubated for 2 hours. After washing four times, chromogenic substrate solution, 2,2'-Azino-bis(3-ethylbenzothiazoline-6- sulfonic acid) diammonium salt (Sigma) in phosphate-citrate buffer (57 μM citric acid, 86 μM $Na_2HPO_4$, 0.6% $H_2O_2$, pH 4.2) was added. The optical density (OD) at 420 nm was measured after about 5 min using a microtitre plate reader. The OD of each sample was standardised using the following formula:

Std OD sample = (OD sample–mean OD negative control)/(mean OD positive control–mean OD negative control)

## Immunization of rabbit and antibody response

Six male, eight-month-old New Zealand white rabbits were purchased from a nearby farm. The rabbits were housed individually with *ad libitum* to drinking water and feed, commercial high protein pellets. After adapting to lab housing for two weeks, animals were immunized with recombinant TR-S14P5 and TR-S21P2, two animals for each recombinant with a dose of 250 μg plus complete Freund's adjuvant (Sigma). After three weeks, the second immunization was carried out similarly but with incomplete Freund's adjuvant (Sigma). The third and fourth immunizations were carried out with similar doses and intervals between vaccination but with 50 μg Quil A (Superfos Biosector, Denmark) as an adjuvant. Two rabbits were unvaccinated and served as control. All animals were bled before immunization and three weeks after each immunization to obtain serum samples. The protocol for the use of animals in this study was approved by the Animal Ethics of the Indonesian Agency for Agricultural Research and Development (Registration Number: Balitbang/BB Litvet/ Rd A/05.01/2021).

A similar ELISA described previously measured antibody response to the immunization, except that anti-rabbit-IgG-horseradish-peroxidase conjugate (Sigma) was used instead of anti-human-IgG, and the rabbit sera were diluted serially. The titre of the antibody against the repeat epitope was expressed as the reciprocal of the highest serum dilution with OD higher than pre-immunised serum at 1: 100 dilution.

A dot-blot assay was carried out to analyze the recognition of the rabbit sera against the SARS CoV-2 virus. A 2-μl volume of formaldehyde-inactivated-SARS-CoV-2-virus suspension, containing approximately 2000 PFU (plaque forming units), was spotted on a nitrocellulose membrane. The inactivated virus was a kind gift from Pt. Biotis Pharmaceutical Indonesia. The membrane was blocked for 2 hr at room temperature in 0.2% non-fat skim milk. Relevant rabbit antiserum diluted 1: 200 in PBS was added and incubated for 2 hr at room temperature. After washing four times in PBST, anti-rabbit-IgG-HRP conjugate (Sigma) was added and incubated for 2 hr at room temperature. After washing four times in PBST, the membrane was incubated in 3,3'-Diaminobenzidine tetrahydrochloride (DAB) solution to expose specific binding of antibodies to the SARS CoV-2.

## Statistical analysis

Results were presented with descriptive statistics. Discriminating power, sensitivity and specificity of ELISA were obtained from ROC curve analysis using IBM SPSS® Statistics.

## Result

Based on amino acid composition, the calculated molecular weights of TR-S14P5 and TR-S21P2 were 13.4 and 13.8 kDa, respectively (Fig 1A). Bacterial cells transformed with genes designed for the tandem repeat epitopes successfully expressed the recombinant

proteins. This was evident from the appearance of a single protein band in SDS PAGE in samples prepared from the IPTG-induced culture of transformed cells. A similar protein band did not appear in similar culture before induction (Fig 1B). The molecular weight of both recombinant proteins was around 17 kDa, slightly higher than those calculated from the amino acid composition.

Both poly histidine-tag-recombinant proteins were bound to nickel sepharose; therefore, the recombinant proteins could be isolated using Ni-NTA column chromatography. The yields of purified proteins were higher when the bacterial lysates were solubilized in denaturing guanidium buffer than those solubilized in the native buffer. This was unexpected since both recombinant proteins consist of mostly polar amino acids, with grand averages of hydrophathicity (GRAVY) of -1.016 for TR-S14P5 and -0.825 for TR-S21P2 (https://web.expasy.org/protparam/). However, recombinant proteins bound to the column were efficiently eluted with the polar native buffer containing 0.5 M imidazole. This indicates that both recombinant proteins were water-soluble in purified form. With this purification method, 58 mg TR-S14P5 and 46 mg TR-21P2 were obtained per litre culture, each with >95 purity (Fig 1C).

## Antibody ELISA

Antibody ELISA indicates that both TR-S14P5 and TR-S21P2 were recognized specifically by sera from patients who suffered from SARS-CoV-2 infections but not by those from non-infected individuals. Sera from non-infected individuals had a low OD, suggesting the specificity of TR-S14P5 or TR-S21P2 for COVID-19, and the interference of contaminating proteins in the TR-S14P5 or TR-S21P2 suspension was insignificant. In either TR-S14P5- or TR-S21P2-based ELISA, the mean std-OD of SARS-CoV2-positive sera was about four times as high as that of negative sera (Fig 2).

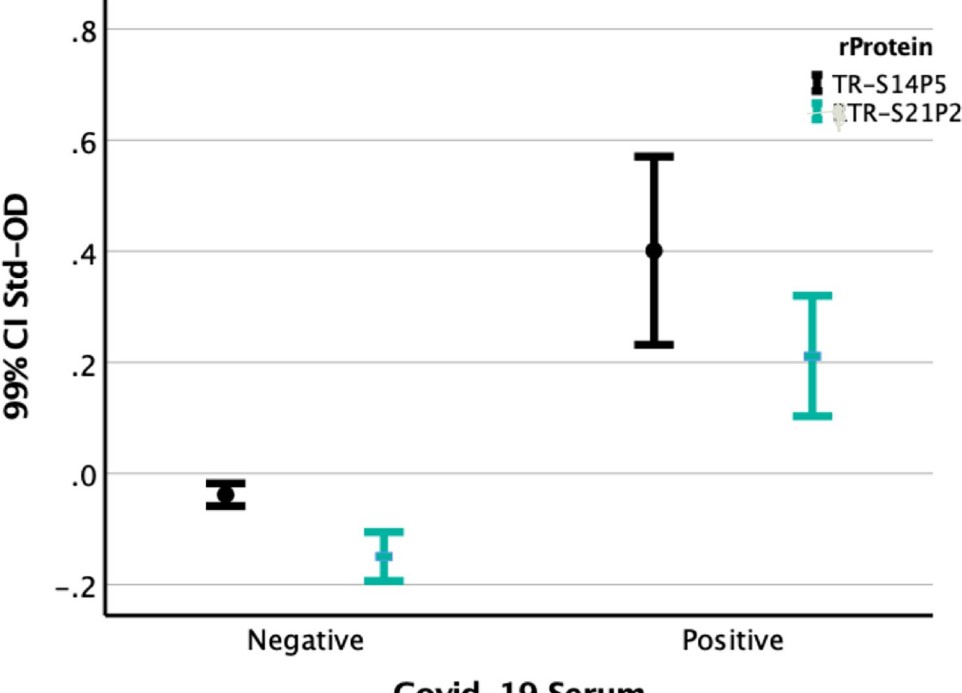

**Fig 2. Differences in the level of antibodies to S14P5 or to S21P2 in sera from SARS CoV-2 infected and those from non-infected persons.**

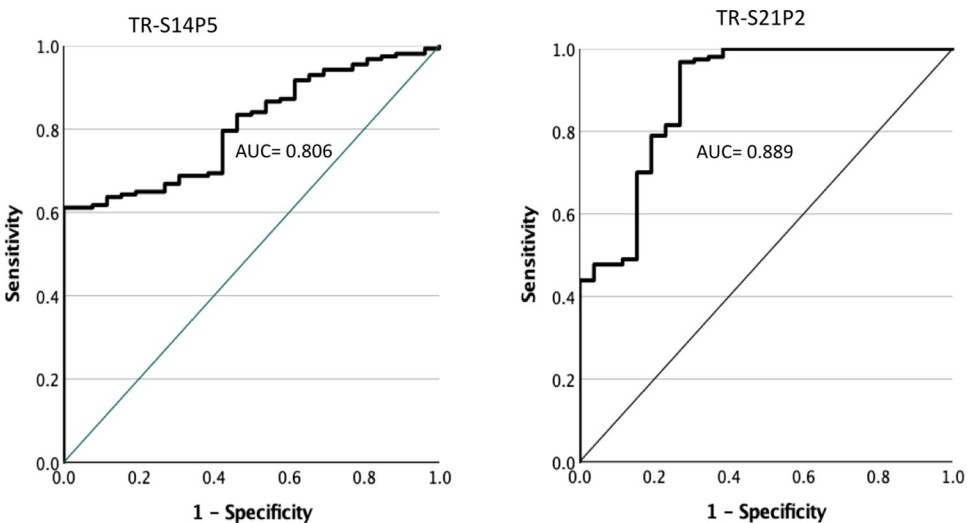

**Fig 3. The accuracy of ELISA using either recombinant TR-S14P5 or TR-S21P2 as the coating antigen in discriminating positive from negative COVID-19 sera.** The area under the curve (AUC) measures discriminating power. Diagnostic tests with an AUC of 0.8–0.9 are considered to have excellent discriminating power.

The discriminating powers TR-S14P5- and TR-S21P2-based ELISAs, as indicated by the AUC (area under the curve), were 0.806 and 0.889, respectively (Fig 3). The AUCs of both ELISA were not different statistically ($P > 0.05$). The S14P5 ELISA was superior in diagnostic specificity compared to the S21P2 ELISA. At the lowest cut-off std OD value, 0.022, which yielded 100% specificity, the S14P5-based ELISA had a sensitivity of 61%, whereas the S21P2-based ELISA, it was only 44% (Table 1). The S21P2-based ELISA, on the other hand, was superior in diagnostic sensitivity. At the lowest cut-off value, -0,136, which yielded 95% sensitivity, the S21P2-based ELISA had a specificity of 73%. In contrast, the S14P5-based ELISA had a specificity of only 23% for the same sensitivity. When both ELISAs were used simultaneously, and the lowest std OD that produced 100% specificity for each ELISA was used as the cut-off, the sensitivity of testing increased to 70% (Table 2).

## Immunogenicity of TR-S14P5 and TR-S21P2

Both TR-S14P5 and TR-S21P2 were highly immunogenic. Two weeks after the first immunization, specific antibodies had risen to a titer above 1: 1000 (Fig 4). The second immunization caused a remarkable increase in titers, to more than ten times as high as those of the first

**Table 1. Diagnostic sensitivity and specificity of TR-S14P5- and TR-S21P2-based ELISA at different standardized-OD cut-offs.**

| TR-S14P5-based ELISA | | | TR-S21P2-based ELISA | | |
|---|---|---|---|---|---|
| Std-OD cut-off | Sensitivity | Specificity | Std-OD cut-off | Sensitivity | Specificity |
| -0,072 | 95% | 23% | -0,166 | 100% | 62% |
| -0,052 | 85% | 46% | -0,136 | 95% | 73% |
| -0,039 | 80% | 54% | -0,116 | 90% | 73% |
| -0,030 | 75% | 58% | -0,096 | 85% | 73% |
| -0,019 | 70% | 58% | -0,089 | 80% | 77% |
| 0,003 | 65% | 81% | -0,084 | 75% | 81% |
| 0,005 | 64% | 85% | -0,073 | 70% | 81% |
| 0,022 | 61% | 100% | 0,024 | 44% | 100% |

**Table 2. Sensitivity and specificity of combined TR-S14P5- and TR-S21P2-based ELISAs at an std-OD cut-off of 0,022 and 0,024, respectively.**

|  | SARS CoV2 infected | SARS CoV2 non-infected | Total |
|---|---|---|---|
| Positive either TR-S14P5- or TR-S21P2 based ELISAs | 110 | 0 | 110 |
| Negative both TR-S14P5- and TR-S21P2 based ELISAs | 47 | 26 | 73 |
| Total | 157 | 26 | 183 |

Sensitivity: 110/157 x 100% = 70%

Specificity: 26/26 x 100% = 100%

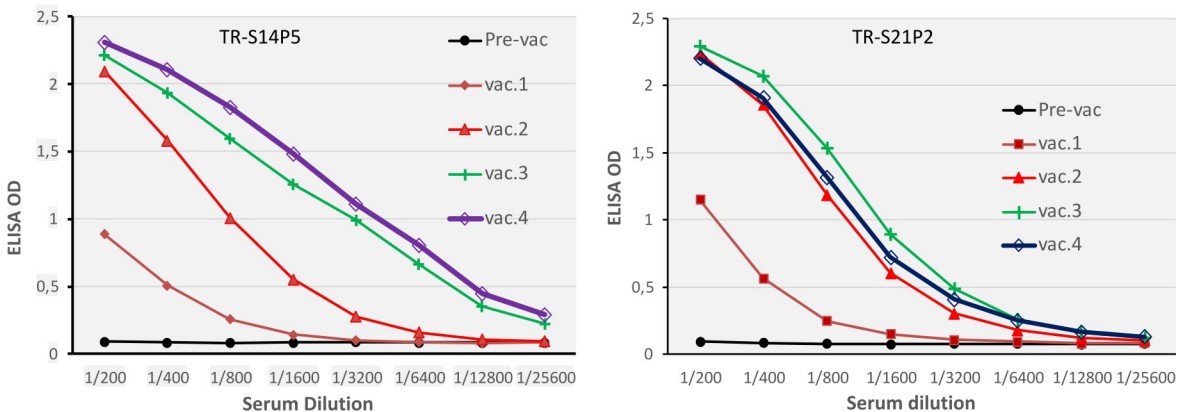

**Fig 4.** Antibody (IgG) responses in rabbits immunized with TR-S14P5 (left) or TR-S21P2 (right). Two rabbits were immunized with each recombinant. Immunizations were carried out four times with three-week intervals between vaccination, with a dose of 250 µg, and with complete Freund's, incomplete Freund's, Quil A, and Quil A adjuvant for the first, second, third and fourth immunizations.

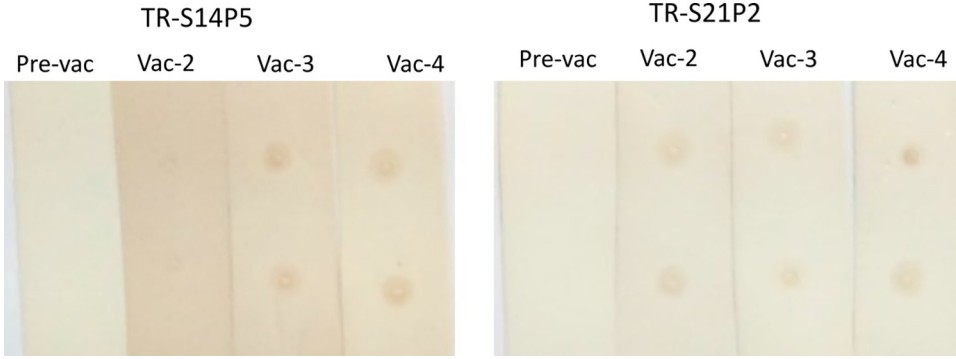

**Fig 5. Dot-blot assay to show specific recognition of formaldehyde-inactivated SARS CoV-2 (SARS-CoV-2/human/IDN/RSDS-RCVTD-UNAIR-35-B/2020) by sera from rabbits before and after the second third and fourth immunization with recombinant TR-S14P5 or TR-21P2.**

immunization. A further increase in the titer was still observed after the third immunization with the TR-S14P5 but to a lower degree with the TR-S21P2. The fourth immunization did not or resulted in only a slight increase in the antibody titers.

The sera from rabbits immunized with the TR-S14P5 or TR-S21P2 recognized specifically SARS CoV-2 virus. Inactivated SARS CoV-2 virus immobilized on nitrocellulose membrane recognized convincingly by sera from rabbits after the third and fourth immunization (Fig 5).

Sera from rabbits prior to immunization did not recognize the virus. Although antibody titers to immunizing TR-S14P5 or TR-S21P2 were remarkable after the first and second immunization, the recognition of inactivated virus was invisible by first, or faintly by the second immunized sera.

## Discussion

This study generated two linear epitopes in the form of tandem repeat (TR-S14P5 and TR-S21P2) expressed in *E. coli*. Functionally as expected, both repeat epitopes recognized by sera from patients confirmed infected by the SARS CoV-2 virus, and sera from rabbits immunized by the proteins recognized specifically SARS-CoV-2 virus. The intended use of the recombinant epitopes is three folds as antigens for antibody detection, to raise antibodies in animals for virus detection, and vaccines against COVID-19.

Linear epitopes, rather than conformational ones, were chosen, despite the latter being much more abundant in the spike protein, because of the possibility of designing them in the form of tandem repeat and expressing them efficiently in *E. coli* with the resulting proteins resistant to denaturation [21]. The linear epitopes S14P5 and S21P2 were small, < 2.4 kDa, peptides; thence, constructing them in tandem repeat with peptide linker in-between enlarged the resulting proteins to sizes that facilitated their production and purification.

To be used as a coating antigen in ELISA, the epitope in the tandem-repeat form has many advantages over the single-peptide form because the tandem repeat is easier to immobilize on the plate and contains more antigenic sites. Consequently, the resulting ELISA should be more sensitive. Previous studies show that immobilization of S14P5 and S21P2 was impractical and costly because peptides were first biotinylated before applying to avidin-coated plates [13, 22].

Based on the area under curves (AUC), which were higher than 0.8, the TR-S14P5- or TR-S21P2-based ELISAs developed in the present study have excellent discriminating power to diagnose COVID-19 positive based on the presence of antibodies to TR-S14P5- or TR-S21P2 [23]. At a specificity of 100%, the ELISA had only 70% sensitivity. However, the actual sensitivity of the test should have been higher if the sample had been collected from patients two weeks or later after the onset of symptoms, as the median time to seroconversion in COVID-19 was reported to be about two weeks [24]. The classification of serum samples into positive or negative for COVID-19 in this study was based on qPCR examination of swab samples from patients with varying durations of infection. Some of the positive samples were at the early stage of infection, most likely to have not been seroconverted. Classifying samples based on the onset of symptoms in the present study was unreliable because many patients did not know when the clinical symptoms began to appear. The sensitivity of the ELISAs would have been higher if the early-stage samples had been excluded from the examination. This assumption was supported by a previous study by Amrun *et al.*, in which analysis was based on serum samples collected at the median post symptoms of 23 days [22]. In that study, the S14P5 ELISA produced 91% sensitivity and 97% specificity, and S21P2 ELISA produced 81% sensitivity and 97%. Another study with a more significant number of samples reported a positivity rate or sensitivity of 77% for S14P5 ELISA and 73% for S21P2 ELISA, or 88.5% for combined S14P5-S21P2 ELISA. Those positivity rates were satisfactory, considering the analysis was based on a broad spectrum of disease severity, from asymptomatic to severe [14].

As coating antigens for ELISA or other immunoassays, the TR-S14P5 and TR-S21P2 have many advantages compared to the biotinylated peptide epitope or recombinant spike or RBD SARS CoV-2 protein. The biotinylated peptide epitopes are obtained commercially and require avidin or streptavidin as anchors for their immobilization on the plate. Recombinant Spike or RBD SARS CoV-2 are expensive because they must be expressed in eukaryotic cells to

glycosylate properly. The TR-S14P5 and S21P2, which are expressed in *E. coli*, are more accessible, faster, and less expensive to produce.

In addition to their use in the antibody-detection immunoassays, the TR-S14P5 and S21P2 could also be used to generate specific antibodies in animals. As shown in this study, the antibodies recognized specifically a local isolate of the SARS-CoV-2 virus. This supports that tandem repeat epitopes function correctly, and the antibodies should be useful for developing an immunoassay for virus or antigen detection. The antibodies should have a strong avidity and high specificity because they are induced by a highly immunogenic, repeat, single epitope.

The initial step in antibody production is the activation of B-cells by cross-linking the cell's receptors by immunogen. The extent of cross-linking determines the speed and magnitude of antibody production. Immunogens with a high density of repeated epitopes are highly effective in cross-linking the immunoglobulin receptor in B-cell [25, 26]. This phenomenon has been used to enhance the immunogenicity of a linear epitope by constructing a new compound, a repeated structure of the linear epitope.

Linear epitopes that can induce neutralizing antibodies have received the most interest to be formulated in repeated form. Several approaches to constructing the repeated epitope have been introduced. Covalent conjugation of peptide epitope to a carrier protein seems to be the earliest procedure. Multiple repeats of a linear epitope amino acid 9–21 of herpes simplex virus type 1 glycoprotein D, known to elicit neutralizing antibody, were synthesized chemically and conjugated covalently to bovine serum albumin. These multi-epitope conjugates evoked neutralizing antibodies in mice that protected the animals against virus challenges [27]. Despite the generated repeated epitope functioning as expected, the protocol, which involves synthesizing peptides and conjugation to BSA, is laborious and costly. Production of the repeated epitope using recombinant technologies seems feasible. Zheng *et al.* constructed a gene for a tandem repeat of amino acid residue 8–23 of glycoprotein-D herpes simplex virus type 1 glycoprotein D and fused it to the amino terminus beta-galactosidase gene, and inserted the fused gene into an adenovirus vector. Mice immunized with the recombinant adenovirus produced neutralizing and protective antibodies, the titer of which correlated positively with the number of repeats. The mean titers of neutralizing Ab induced by 1, 2, 3 and 4 repeats were 64, 230, 314 and 448, respectively. A titer of at least 80 protected mice from death after a challenge with a heterologous human herpes simplex virus [28].

Production of repeat epitope with recombinant adenovirus is beneficial when the linear epitope is a glycopeptide. When glycosylation is unnecessary, producing repeated linear epitopes using the *E. coli* expression system is more practical. The superiority of the *E. coli* expression system is exemplified by the production of tandem repeat M2e, the external domain of influenza-virus M2 protein. The 20 amino-acid-long M2e potentially develops into a universal influenza vaccine because it elicits protective antibodies and is conserved among type-A-influenza viruses. The need for a universal vaccine is enormous because conventional influenza vaccines are based on the virus surface glycoproteins, which are sub-type specific and rapidly mutated. The M2e has low immunogenicity, but this problem has been solved by constructing the peptide in tandem repeat and expressed in *E. coli.* Liu *et al.* constructed recombinant protein consisting of 2, 4, 8, or 16 tandem-repeat of M2e, fused to glutathione-S-transferase (GST), inserted into a pGEX-4T-2 plasmid, and expressed *in E. coli.* When immunized rabbits were challenged with the influenza virus, it was found that the degree of protection was correlated with the number of M2e repeats. Rabbits immunized with 16 M2e repeats were 100% survival, those with four repeats were 50% survival, and those with single M2e were 0% survival [29].

Higher repeats of M2e are suggested to be more effective in cross-linking the Ig-receptor on the immature B-cells, which results in a higher level of protective antibody [30]. Using the

surface plasmon resonance (SPR) technique, the authors calculated that the polyclonal antibody induced by 4 x M2e-repeat fusion protein has an average affinity constant (KA) of 5.3 x $10^8$ M-1, which was nearly 200 times as high as that induced by the single M2e fusion protein.

Constructing a repeat epitope to enhance the immunogenicity of the epitope is very useful in vaccine development. Epitope 10E8, located in the membrane-proximal external region (MPER) of the gp41 protein human immunodeficiency virus 1 (HIV 1), is a good example. This epitope is involved in viral-host fusion and can induce neutralizing antibodies. However, this epitope has very weak immunogenicity because it tends to insert into the host cell membrane, thereby avoiding the binding with the Ig-receptor of the B-cells [31]. These authors successfully enhanced the immunogenicity of epitope 10E8 by synthesizing a four tandem repeat of the epitope using standard solid-phase 9-fluorenyl-methoxycarbonyl (FMOC). Rabbits immunized with the tandem repeat epitope elicited antibodies with neutralizing activity and antibody-dependent cell-mediated cytotoxicity (ADCC) against HIV-1, whereas those immunized with single or non-repeat epitope 10E8 did not. The structure of the tandem repeat 10E8 was predicted to have a well-arranged tandem helical conformation. This conformation allows some of the epitopes to be exposed outside the lipid membrane and hence readily recognized by B-cell receptors [31].

Since linear epitopes S14P5 and S21P2 elicit neutralizing antibodies, they should potentially develop into vaccines. Poh et al. found that neutralizing activities of COVID-19- convalescent sera were reduced after depletion with the S14P5 or S21P2 synthetic peptide [13]. This finding was in line with a subsequent study showing a significant correlation between the serum neutralization titer and antibody titers to S14P5 ($p<0.01$, rs = 0.510) and antibody to S21P2 ($p<0.05$, rs = 0.222) [14]. Those correlations are considered close because the correlation between the serum neutralization titer with antibody titers to RBD was only 0.602 [14]. The exact mechanism for neutralizing activities is unknown; however, since the S14P5 epitope is in the vicinity of RBD, the elicited antibody may act as a steric hindrance. The S21P2 is part of fusion peptides, and the elicit antibody may inhibit the fusion of the viral-host membrane [13]. Unfortunately, the current study did not measure the neutralizing activity of sera from rabbits immunized with the TR-S14P5 or TR-S21P2. Further study should be conducted to obtain this critical information.

The tandem repeat of either one or both has the potential to develop into effective vaccines. The TR-S14P5 and TR-S21P2 were proper immunologically since sera from COVID-19 patients recognized them, and antibodies against them recognized SARS CoV-2 virus. Both epitopes were highly conserved and easy to produce. As linear epitopes, the TR-S14P5 and TR-S21P2 are denaturation resistant, the quality of which is ideal for mucosal or intra-nasal vaccines. Most, if not all, epitopes in the RBD regions inducing neutralizing antibodies are conformational epitopes, which are susceptible to denaturation [32]. The development of mucosal vaccine from linear epitope was demonstrated in HIV-1.

Linear epitope ELDKWA, located in gp41, elicits neutralising antibody by inhibiting viral transcytosis, has been formulated in tandem repeat and fused with immunomodulating protein, CTB (B-sub-unit cholera toxin) and expressed in *E. coli*. Mice immunised intranasally with the recombinant protein elicited neutralising nasal IgA and systemic IgG [33]. A similar approach for developing an intra-nasal vaccine using TR-S14P5 or S21P2 may be applicable.

In summary, the linear epitope S15P5 and S21P2 reported previously elicited neutralizing antibodies were formulated in tandem repeat and expressed in *E. coli*. The purified TR-S15P5 and S21P2 can be used as coating antigens in indirect ELISA that are potentially used as a serological test for COVID-19. The determining power, indicated by the area under the curve (AUC), of the TR-S15P5- and S21P2-based ELISA on serum samples collected from COVID-19-positive and negative-confirmed individuals were 0.806 and 0.889, respectively. The

purified TR-S15P5 and S21P2 were highly immunogenic, and antibodies induced by each protein recognized SARS-CoV-2 virus. The antibodies, therefore, could be developed into a rapid test for the detection of SARS-CoV-2 antigen. In addition to their use as a diagnostic test, the TR-S15P5 and S21P2 also have a great potential to be used as vaccines. The TR-S15P5 and S21P2 are inexpensively produced, highly immunogenic and denaturation resistant; thence, they are ideal candidates for intranasal vaccines.

## Acknowledgments

The authors thank Mrs Gita Sekarmila, Mr Achpas and the animal caretakers for their excellent technical assistance.

## Author Contributions

**Conceptualization:** Simson Tarigan.

**Data curation:** Simson Tarigan, Dianita Sugiartanti, Ryandini Putri, Andriani, Harimurti Nuradji, Marthino Robinson, Niniek Wiendayanthi.

**Formal analysis:** Simson Tarigan.

**Funding acquisition:** N. L. P. Indi Dharmayanti, Fadjry Djufri.

**Investigation:** Simson Tarigan, Dianita Sugiartanti, Ryandini Putri, Andriani, Marthino Robinson, Niniek Wiendayanthi.

**Methodology:** Simson Tarigan.

**Project administration:** Simson Tarigan, Andriani, Harimurti Nuradji.

**Supervision:** N. L. P. Indi Dharmayanti, Fadjry Djufri.

**Validation:** Simson Tarigan.

**Visualization:** Simson Tarigan.

**Writing – original draft:** Simson Tarigan.

**Writing – review & editing:** Simson Tarigan.

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
