## [Decision Letter · Decision Letter 0]

18 Oct 2022

PONE-D-22-17597Characterization of Two Linear Epitopes SARS CoV-2 Spike Protein Formulated in Tandem RepeatPLOS ONE

Dear Dr. Tarigan,

Thank you for submitting your manuscript to PLOS ONE. After careful consideration, we feel that it has merit but does not fully meet PLOS ONE’s publication criteria as it currently stands. Therefore, we invite you to submit a revised version of the manuscript that addresses the points raised during the review process.

We look forward to receiving your revised manuscript.

Kind regards,

Faten Abdelaal Okda

Academic Editor

PLOS ONE

Journal Requirements:

Reviewers' comments:

Reviewer's Responses to Questions

**Comments to the Author**

1. Is the manuscript technically sound, and do the data support the conclusions?

Reviewer #1: Yes

Reviewer #2: Yes

Reviewer #3: Partly

2. Has the statistical analysis been performed appropriately and rigorously? 

Reviewer #1: Yes

Reviewer #2: Yes

Reviewer #3: I Don't Know

3. Have the authors made all data underlying the findings in their manuscript fully available?

Reviewer #1: Yes

Reviewer #2: Yes

Reviewer #3: Yes

4. Is the manuscript presented in an intelligible fashion and written in standard English?

Reviewer #1: Yes

Reviewer #2: No

Reviewer #3: Yes

5. Review Comments to the Author

Reviewer #1: 1.The work is of great practical significance. It is suggested that commercial antigen (linear epitope) should be added as a positive control to detect COVID-19 convalescent patient's serum as coated antigen.

2.S14P5 and S21P2 are linear epitopes on RBD. Considering the rapid variation of Covid-19, especially the S protein, it is suggested to introduce the conservativeness of these two epitopes in the Introduction section.

3.The purity of some FIG 1C and proteins was > 95% using SDA-PAGE in Fig1 c. Was it observed by naked eyes or other methods? The figure shows that there are many miscellaneous bands. If it is used as a coated antigen, does this purity meet the requirements? It is suggested to continue to optimize the purification conditions.

4.S14P5 and S21P2, which are the epitopes of neutralizing antibodies, are used in the sera of immunized rabbits, but they only say that they can recognize SARS-CoV-2 virus. It is suggested to detect the titer of neutralizing antibodies in immunized sera.

5.The grammar of the article is formal and easy to understand, but there are still some minor mistakes, such as Line 43, "first identified in December 2010". Is 2010 a clerical error? Line 133 and 149, writing format of chemical formula, etc.; It is recommended to check carefully.

Reviewer #2: The authors present initial data on a potential SARS-2 vaccine based on linear epitopes within the spike protein. The data presented support that the linear epitope subunit selected is immunogenic. Such an approach is scalable and safe.

The manuscript needs substantial revision and editing. It is brief and does not include efficacy data.

The authors need to clearly state the purpose and rationale-why do we need another vaccine and how is the one proposed an improvement of mRNA vaccines, killed vaccines, or other subunit vaccines? How does the vaccine perform against circulating SARS-2 strains?

The abstract contains many details that are better suited for the body of the manuscript, particularly lines 26-27.

Materials and Methods are lengthy and thorough, however the paper would be more engaging if many of the methods were shortened and included more references. Since the peptides are generated from bacteria (E. coli) endotoxin testing should be included.

Is a scrambled peptide necessary to show specificity?

Need consistency between Covid or covid throughout. COVID is commonly accepted.

The manuscript needs to more thoroughly referenced, the authors depend on reviews rather than primary references. Example lines 47, 48-62, 90-117, and 369-374.

The data that the subunit vaccine is immunogenic is convincing, the manuscript would be improved if tested against multiple variants.

There are many typos and sentence structures that can be greatly improved by proofing by a English proofing service.

Line 43, 2010 or 2019?

Line 66 "thereby" preventing infection

Line 178 "skim" not "skimmed"

Line 261 and instead of "dan"

Line 302 use positive or negative to describe the samples.

Reviewer #3: Characterization of Two Linear Epitopes 1 SARS CoV-2 Spike Protein Formulated in Tandem Repeat

Summary:

In the present study, authors generated tandem repeats (TR-S145 and TR-S21P2) from previously identified linear neutralizing epitopes. TR-S145 and TR-S21P2 were expressed in E.coli and purified using Ni-affinity chromatography with yields of 58 mg/L and 46 mg/L respectively. TR-S14P5 (AUC- 0.806) or TR-S21P2 (AUC-0.889) based ELISA showed good discrimination among sera samples collected from Covid-19-positive and negative-confirmed individuals. The S14P5 ELISA was superior in diagnostic specificity compared to the S21P2 ELISA. On the other hand, the S21P2-based ELISA was superior in diagnostic sensitivity. When S14P5 and S21P2 ELISA were combined, at a specificity of 100%, the ELISA had only 70% sensitivity. Sera from rabbits after 3rd and 4th immunization recognized inactivated SARS-CoV-2 virus, immobilized on nitrocellulose membrane.

Remarks:

• Authors have claimed purified TR-S15P5 and S21P2 can be used as coating antigens in indirect ELISA that are potentially used as a serological test for Covid-19. As coating antigens for ELISA, the TR-S14P5 and S21P2 would have advantage over linear epitope peptides because a tandem repeat is easier to immobilize and contain more antigenic sites.The sensitivity and specificity of TR-S15P5 and S21P2-based ELISA reported here is lower than the linear epitope-based ELISA. Both S14P5 and S21P2 have a moderate specificity and sensitivity level of >80%, at 10 days post illness-onset. However, at median 23 days post illness-onset, the percentage recognition for all is >95% (Amrun et. al).

• Four immunizations with a very high dose, 250μg of TR-S14P5- and TR-S21P2, elicited humoral immune response. Only Dot Blot, a qualitative assay, was performed to show that elicited antibodies could recognize SARS-CoV-2 virus. Sera only after 3rd and fourth immunization convincingly recognized inactivated SARS-CoV-2 virus. These tandem repeats were derived from epitopes eliciting neutralizing antibodies, but no neutralization assay was performed in the study.

Major issues

The authors suggest these peptides can be used for either diagnostic or vaccine purposes. The sensitivity for diagnostic applications is not high. Also at this point in time, the value of serological diagnostics is unclear when a large fraction of the world’s population has either been infected or vaccinated. Perhaps this might be useful for monitoring duration of immune responses, but then results with the peptides need to be compared with full spike, RBD or nucleocapsid. For vaccines, despite the initial Nature Comm paper of Poh et al, there is little evidence that these two linear peptides would be useful for a vaccine. The amount of depletion of neutralizing activity seen in the Poh et al paper is very marginal.

Overall, there does not seem to be sufficient novelty or important results to justify publication

Minor points and corrections

1) Generally, the working concentration of Kanamycin is50 μg/ml. Why have authors used 30μg/ml as working concentration?

2) For purification, why two different denaturants, 6M Gdm-Cl (lysis buffer) and 8M Urea (Wash buffer) was used?

3) Why were different adjuvants, complete Freund’s adjuvant (1st immunization), incomplete Freund’s adjuvant (2nd immunization) and Quil A (for 3rd and 4th immunization) were used?

Corrections:

• Introduction (Line: 42-44):

Within 30 months since first identified in December 2020

Within 30 months since first identified in December 2010, the SARS CoV-2 virus has infected at least 516,476,402 people globally, including 6,258,023, resulting in death.

• Introduction (Line: 61-62):

Remove additional Parentheses Zhu et al. [1]

All parts of S2 involve in the host-viral membrane fusion. In addition to membrane fusion, TM and CT play vital roles in spike protein trimerisation and anchoring the trimeric spike proteins, review by Zhu et al. [1])

• Introduction (Line: 72-73):

S21P2 is in the S2 subunit close to the fusion peptide

Epitope S14P5 is located in the S1 subunit downstream of RBD, whereas epitope S21P2 is in the S2 subunit closed to the fusion peptide.

• Introduction (Line: 79-80):

E. coli should be in italics

In the present study, the S14P5 and S21P2 epitopes were formulated in tandem repeat and expressed in E. coli to increase their immunogenicity and ease of production.

• Materials and methods (Line: 92-93):

Shouldn’t it be “placed”

The TR-S14P and TR-21P2 were constructed by four tandem repeat epitopes, S14P5 or S21P4, and a flexible peptide linker (GGGS) was paced between the epitopes (Fig 1A).

• Materials and methods (Line: 109-110):

A preliminary experiment

A Preliminary experiment indicated that the hybrid methods were the most efficient for both recombinants.

• Materials and methods (Line: 132):

TR-S21P2

The recombinant protein,TR-S14P5) or TR-S21P, was dissolved in 0.1 M carbonate buffer, pH 9.6 at 20 μg/ml, 100 μl of the suspension was added to each well of a maxisorb Nunc plate then incubated at 4oC overnight.

• Materials and methods (Line: 136):

Remove additional Parentheses non-fat skim milk (2% (w/v)

non-fat skim milk (2% (w/v))

• Materials and methods (Line: 139):

Subscripts should be used for chemical formulas of salts: 8 μM Na2HPO4, 1.5 μM KH2PO4

stringent washing buffer [640 μM NaCl, 3 μM KCl, 8 μM Na2HPO4, 1.5 μM KH2PO4,

• Materials and methods (Line: 144, 172):

Subscripts should be used for chemical formulas of salts: 86μM Na2HPO4 , 0.6%(v/v) H2O2

[57 μM citric acid, 86 μM Na2HPO4, 0.6%(v/v) H2O2, pH 4.2] was added. The optical

• Materials and methods (Line: 165):

TR-S14P5

coated with 2 μg TR-15P5 or TR-S21P2 per well at 4oC overnight. After blocking with 0.2%

• Results (Line: 203):

TR-S14P5

hydropathicity of -0.970 for TR-S4P5 and -0.787 for TR-S21P2

• Results (Line: 207):

TR-S21P2

method, 58 mg TR-S14P5 and 46 mg TR-21P2 were obtained, each with >95 purity (Fig 1C).

• Results (Line: 217-218):

3-to-4-fold

In either TR-S14P5- or TR-S21P2-based ELISA, the mean std-OD of SARS-CoV2-positive sera was about 3 to 4 as high as that of negative sera (Fig 2).

Non-uniform referencing style across text,

Example1:

The presence of epitopes in the RBD elicit neutralising antibodies is undisputable. However,

neutralising antibodies evoked by epitopes in the spike protein located outside the RBD has

72 also been identified (Seydoux et al., 2020).

Example2:

The epitopes peptides synthesised chemically have also been shown to be useful

78 as serological tests [5, 6]

6. PLOS authors have the option to publish the peer review history of their article (what does this mean?). If published, this will include your full peer review and any attached files.

Reviewer #1: **Yes: **Shengli Meng

Reviewer #2: No

Reviewer #3: No

---

## [Author Response · Author response to Decision Letter 0]

24 Dec 2022

Response to the Reviewers

Manuscript # PONE-D-22-17597 “Two Linear Epitopes SARS CoV-2 Spike Protein Formulated in Tandem Repeat" 

Reviewer #1: 

1.The work is of great practical signiﬁcance. It is suggested that commercial antigen (linear epitope) should be added as a positive control to detect COVID-19 convalescent patient's serum as coated antigen.

We appreciate the reviewer for this suggestion. Running the tandem repeat epitopes alongside their counterpart, a non-repeat commercial peptide epitope, would have provided information regarding the increased antigenicity and immunogenicity of the repeated structure. However, S14P5 or S21P2 peptides most likely bind poorly to ordinary polystyrene plates unless they were biotinylated and immobilized on a streptavidin-coated plate, as shown in previous studies. In addition, synthetic peptides without conjugation to a carrier protein usually have poor immunogenicity. Therefore, synthetic S14P5 or S21P2 were not used as controls for TR-S14P5 and TR21P2 in this study.

2.S14P5 and S21P2 are linear epitopes on RBD. Considering the rapid variation of Covid-19, especially the S protein, it is suggested to introduce the conservativeness of these two epitopes in the Introduction section.

Yes, this suggestion is importance. Addressing this suggestion, we add two paragraphs in the Introduction (line 66-79): 

“However, while the implementation of vaccinations was still in full swing, a new variant of concern (VOC) SARS CoV2 emerged and circulated rapidly in succession, Alpha, Beta, Gamma, Delta and Omicron variants. The vaccine's effectiveness against those variants decreased progressively, from a slight decrease against the Alpha variant to a practically ineffective against the Omicron variant [4, 5]. The time from the first report of SARS CoV-2 to the emergence of the Omicron variant was only about two years [6-9]. This means that mutations leading to immune evasion in SARS-CoV-2 occur rapidly.

The ineffectiveness of the vaccines was associated with the fact that the vaccines rely on spike protein which is highly prone to mutation [10]. Guruprasad recently analysed the amino acid sequence of 303,250 spike proteins deposited in the NCBI virus database on 2 November 2021 and found that 96.5% of the proteins had mutated to various degrees. The domains of the protein were not equally susceptible to mutation; the N terminal (residues 1-302), the receptor binding (333-327), and the S1D (residues 594-674) domains were the most frequently to undergo mutations [11].”

3.The purity of some FIG 1C and proteins was > 95% using SDA-PAGE in Fig1 c. Was it observed by naked eyes or other methods? The ﬁgure shows that there are many miscellaneous bands. If it is used as a coated antigen, does this purity meet the requirements? It is suggested to continue to optimize the puriﬁcation conditions.

Thank you for these suggestion. The purity of the protein on SDS PAGE was judged by naked eyes. We think that the purity should meet the requirement as coating plate in ELISA. To address this suggestion we add “Assessment of purity was based on the thickness of the Coomassie-blue-stained target protein band relative to the contaminant bands. In Materials and Method, line 138-139, and “Assessment of purity was based on the thickness of the Coomassie-blue-stained target protein band relative to the contaminant bands.” in the Results section, line 235-237.

4.S14P5 and S21P2, which are the epitopes of neutralizing antibodies, are used in the sera of immunized rabbits, but they only say that they can recognize SARS-CoV-2 virus. It is suggested to detect the titer of neutralizing antibodies in immunized sera.

We agree with the reviewer on the importance of measuring the neutralizing activity of sera from rabbits immunized with the TR-S14P5 or TR-S21P2. We did not do the serum neutralization test because our lab was not qualified to do a serum neutralization test for SARS-CoV2. Laboratories in Indonesia that were qualified to do the neutralization tests were too busy then to collaborate with us to carry out the neutralizing test. However, we have obtained a new grant for a further study in which we are performing the neutralization test. Accommodating the reviewer's comments, we have added: "Unfortunately, the current study did not measure the neutralizing activity of sera from rabbits immunized with the TR-S14P5 or TR-S21P2. Further study should be conducted to obtain this critical information." In the Discussion section, lines 420-422.

5.The grammar of the article is formal and easy to understand, but there are still some minor mistakes, such as Line 43, "ﬁrst identiﬁed in December 2010". Is 2010 a clerical error? Line 133 and 149, writing format of chemical formula, etc.; It is recommended to check carefully.

Thank you. The mistakes have been fixed

 

Response to the Reviewers

Manuscript # PONE-D-22-17597 “Two Linear Epitopes SARS CoV-2 Spike Protein Formulated in Tandem Repeat" 

Reviewer #2: 

The authors present initial data on a potential SARS-2 vaccine based on linear epitopes within the spike protein. The data presented support that the linear epitope subunit selected is immunogenic. Such an approach is scalable and safe.

The manuscript needs substantial revision and editing. It is brief and does not include efﬁcacy data.

• We appreciate the reviewer’s comments. 

• As seen from the file with track changes, we have made substantial revisions to the manuscript. 

• Efficacy data, unfortunately, did not available because we did not carry out any vaccination-trial experiment. A vaccination-trial experiment was beyond the scope of the current study. 

The authors need to clearly state the purpose and rationale-why do we need another vaccine and how is the one proposed an improvement of mRNA vaccines, killed vaccines, or other subunit vaccines? How does the vaccine perform against circulating SARS-2 strains?

We appreciate the comments. To accommodate those important suggestions, we have rewritten the Introduction and accommodated all reviewer’s suggestions. 

….” mutations leading to immune evasion in SARS-CoV-2 occur rapidly” (lines 70) ……. “updating vaccines too frequently is not feasible. An alternative for maintaining vaccine effectiveness is to generate vaccines from conserved epitopes of the virus” (lines 81-83) …...…….. “two conserved, linear epitopes, S15P5 and S21P2, in the spike protein eliciting neutralising antibodies” (lines 84-85)……. “To be used as vaccines, however, the epitopes require some manipulation to increase their capacity to induce immune responses and to ease their production. Without structural modification, the immune responses elicited by the epitopes may not adequately inhibit infection. In the present study, the S14P5 and S21P2 epitopes were formulated in tandem repeat and expressed in E. coli to increase their immunogenicity and ease of production” (lines 95-97)…………. “It is expected that vaccines developed from the recombinant S14P5 or S21P2 tandem repeat could complement the current Covid-19 vaccines to overcome the rapidly mutated SARS-COV-2” (lines 106-108).

The abstract contains many details that are better suited for the body of the manuscript, particularly lines 26-27.

We appreciate the comments. We have rewritten the abstract

Materials and Methods are lengthy and thorough, however the paper would be more engaging if many of the methods were shortened and included more references. Since the peptides are generated from bacteria (E. coli) endotoxin testing should be included.

We appreciate the comments.

• The Materials and Methods have been shortened. The ELISA protocol for measuring antibody titers in immunized rabbits has been mostly deleted.

• Ideally, endotoxin assay was carried out since the recombinant proteins were injected into rabbits. However, endotoxin contamination in the purified recombinant protein may not be significant since no signs of endotoxin toxicity observed after the injection. For in vitro use such as ELISA, endotoxin assay was not required 

Is a scrambled peptide necessary to show speciﬁcity?

We apologize for not understanding this question. We have emailed the editor asking for an explanation of this question, but so far, we have not received a reply

Need consistency between Covid or covid throughout. COVID is commonly accepted.

We have replaced all “Covid” to “COVID”

The manuscript needs to more thoroughly referenced, the authors depend on reviews rather than primary references. Example lines 47, 48-62, 90-117, and 369-374.

Thank you. We have relaced all the review references with primay ones.We hope we are not misunderstand the reviewer. Reference in paragraph line 47” (https://covid19.who.int ) is the real time report of COVID 19 by the WHO, so it is primary source. References in 90-117, and 369-374 are primary studies.

The data that the subunit vaccine is immunogenic is convincing, the manuscript would be improved if tested against multiple variants.

We agree with the reviewer that this manuscript would be improved if tested against multiple variants, especially those with mutated S14P5 or S21P2 epitopes. Unfortunately, we did not have access to multiple variants. 

There are many typos and sentence structures that can be greatly improved by prooﬁng by a English prooﬁng service.

Line 43, 2010 or 2019?

Line 66 "thereby" preventing infection Line 178 "skim" no "skimmed"

Line 261 and instead of "dan"

Line 302 use positive or negative to describe the samples.

The typos have been revised. 

If needed we are happy to hire an English proofing service

 

Response to the Reviewers

Manuscript # PONE-D-22-17597 “Two Linear Epitopes SARS CoV-2 Spike Protein Formulated in Tandem Repeat" 

Reviewer #3: 

Major issues

The authors suggest these peptides can be used for either diagnostic or vaccine purposes. The sensitivity for diagnostic applications is not high. Also at this point in time, the value of serological diagnostics is unclear when a large fraction of the world’s population has either been infected or vaccinated. Perhaps this might be useful for monitoring duration of immune responses, but then results with the peptides need to be compared with full spike, RBD or nucleocapsid. For vaccines, despite the initial Nature Comm paper of Poh et al, there is little evidence that these two linear peptides would be useful for a vaccine. The amount of depletion of neutralizing activity seen in the Poh et al paper is very marginal.

Overall, there does not seem to be sufficient novelty or important results to justify publication

We appreciate the reviewer’s comments. After carefully considering this comments and rereading our manuscript, we realised that we have not expressed ourselves clearly enough in our text. To avoid any misunderstandings and to address the reviewer’s concern, we have rewritten the Abstract, Introduction and made a major revisions to the Discussion. 

The low sensitivity of TR-S14P5- or TR-S21P2-based ELISA has been discussed at length in the manuscript. Serum samples derived from early-stage Covid-19 patients were considered to be the leading cause of the low sensitivity. This low sensitivity is unlikely associated with the inherent property of the epitope. In a recent study, Masterson and Sarder (2022) (https://doi.org/10.1021/acsami.2c06599) demonstrated a 100% sensitivity of a serologic test based on the epitopes, even with serum samples from individuals infected with the highly mutated Omicron variant of SARS. Since TR-S14P5 and TR-S21P2 are highly conserved, tests developed from these repeated epitopes might have higher specificity than those developed from a full spike, RBD or nucleocapsid. We envisage that sensitivity and specificity ELISA based on the TR-14P5 or P21P2 can be improved to the extent that it meets the requirements as a monitoring tool for monitoring immune responses resulting from vaccination.

We agree that serological tests at this time are not as necessary as at the early stage of the Covid 19, i.e. before the widespread vaccination. However, serological tests' role in diagnostics may become as important again in the future when vaccination is discontinued.

The amount of depletion of neutralizing activity by S14P5 or P21P2 seen in the Poh et al paper was about 20%. This amount of depletion may not be high enough to warrant S14P5 or P21P2 as a potential vaccine. However, the neutralizing capacity of the epitope may be increased. Formulation of these epitopes in the form of a tandem repeat, as carried out in this study, was an attempt to increase the immunogenicity of the epitopes. Further increase of the immunogenicity may be obtained through the use of a potent adjuvant. 

Interest in these linear epitopes stems from the fact that they are highly conserved, so developed vaccines can be used for various variants of SARS-CoV2. In addition, the epitopes are denaturation-resistant, making them suitable for intranasal vaccines. The epitope-based intranasal vaccines may be used to complement the current Covid-19 vaccines. The complementation is essential because the current vaccines induce only partial protection. 

Minor points and corrections

1) Generally, the working concentration of Kanamycin is 50 μg/ml. Why have authors used 30 μg/ml as working concentration?

The use of 30 µg/mL was based on the protocols provided, pET system manual (https://research.fredhutch.org/content/dam/stripe/hahn/methods/biochem/pet.pdf). Also, we have tried that our non-transformed BL-21 E. coli did not grow at that concentration.

2) For purification, why two different denaturants, 6M Gdm-Cl (lysis buffer) and 8M Urea (Wash buffer) was used?

Again, the use of the two denaturants were based on the protocol provided by producer of the column: Ni-NTA Purification System-For purification of poly histidine-containing recombinant proteins (https://www.thermofisher.com). 

3) Why were different adjuvants, complete Freund’s adjuvant (1st immunization), incomplete Freund’s adjuvant (2nd immunization) and Quil A (for 3rd and 4th immunization) were used?

This is the practice in our laboratory to produce high titre antibody. The use of Quil A in the third and fourth immunization is to reduce use of more toxic. Freund’s adjuvant

Corrections:

• Introduction (Line: 42-44):

Within 30 months since ﬁrst identiﬁed in December 2020

Within 30 months since ﬁrst identiﬁed in December 2010, the SARS CoV-2 virus has infected at least 516,476,402 people globally, including 6,258,023, resulting in death.

Thank you. The errors have been fixed

• Introduction (Line: 61-62):

Remove additional Parentheses Zhu et al. [1]

All parts of S2 involve in the host-viral membrane fusion. In addition to membrane fusion, TM and CT play vital roles in spike protein trimerisation and anchoring the trimeric spike proteins, review by Zhu et al. [1])

Thank you. The errors have been fixed

• Introduction (Line: 72-73):

S21P2 is in the S2 subunit close to the fusion peptide

Epitope S14P5 is located in the S1 subunit downstream of RBD, whereas epitope S21P2 is in the S2 subunit closed to the fusion peptide.

Thank you. The errors have been fixed

• Introduction (Line: 79-80): E. coli should be in italics

In the present study, the S14P5 and S21P2 epitopes were formulated in tandem repeat and expressed in E. coli to increase their immunogenicity and ease of production.

Thank you. The errors have been fixed

• Materials and methods (Line: 92-93): Shouldn’t it be “placed”

The TR-S14P and TR-21P2 were constructed by four tandem repeat epitopes, S14P5 or S21P4, and a ﬂexible peptide linker (GGGS) was paced between the epitopes (Fig 1A).

Thank you. We replaced “placed” with “inserted”

• Materials and methods (Line: 109-110): A preliminary experiment

A Preliminary experiment indicated that the hybrid methods were the most efﬁcient for both recombinants.

Thank you. The error have been fixed. “A Preliminary experiment indicated that the hybrid methods were the most efﬁcient for both recombinants” was delated

• Materials and methods (Line: 132): TR-S21P2 The recombinant protein,TR-S14P5) or TR-S21P, was dissolved in 0.1 M carbonate buffer, pH 9.6 at 20 μg/ml, 100 μl of the suspension was added to each well of a maxisorb Nunc plate then incubated at 4oC overnight.

Thank you. The error fixed by writing the sentence into two sentences

• Materials and methods (Line: 136):

Remove additional Parentheses non-fat skim milk (2% (w/v) non-fat skim milk (2% (w/v))

Thank you. The error have been fixed.

• Materials and methods (Line: 139):

Subscripts should be used for chemical formulas of salts: 8 μM Na2HPO4, 1.5 μM KH2PO4 stringent washing buffer [640 μM NaCl, 3 μM KCl, 8 μM Na2HPO4, 1.5 μM KH2PO4,

Thank you. The errors have been fixed.

• Materials and methods (Line: 144, 172):

Subscripts should be used for chemical formulas of salts: 86μM Na2HPO4 , 0.6%(v/v) H2O2 [57 μM citric acid, 86 μM Na2HPO4, 0.6%(v/v) H2O2, pH 4.2] was added. 

• Materials and methods (Line: 165): TR-S14P5 coated with 2 μg TR-15P5 or TR-S21P2 per well at 4oC overnight. After blocking with 0.2%

Thank you. The errors have been fixed.

• Results (Line:203): TR-S14P5

hydropathicity of -0.970 for TR-S4P5 and -0.787 for TR-S21P2

Thank you. The typos have been fixed.

• Results (Line: 207): TR-S21P2

method, 58 mg TR-S14P5 and 46 mg TR-21P2 were obtained, each with >95 purity (Fig 1C).

Thank you. The errors have been fixed.

• Results (Line: 217-218): 3-to-4-fold

In either TR-S14P5- or TR-S21P2-based ELISA, the mean std-OD of SARS-CoV2-positive sera was about 3 to 4 as high as that of negative sera (Fig 2).

Thank you. The errors have been fixed. “positive sera was about 4 times as high as that of negative sera”

Non-uniform referencing style across text, Example1:

The presence of epitopes in the RBD elicit neutralising antibodies is undisputable. However, neutralising antibodies evoked by epitopes in the spike protein located outside the RBD has 72 also been identiﬁed (Seydoux et al., 2020).

Example2:

The epitopes peptides synthesised chemically have also been shown to be useful 78 as serological tests [5, 6]

Thank you. The errors have been fixed.

---

## [Editor Report · Decision Letter 1]

5 Jan 2023

Characterization of Two Linear Epitopes SARS CoV-2 Spike Protein Formulated in Tandem Repeat

PONE-D-22-17597R1

Dear Dr. Tarigan,

We’re pleased to inform you that your manuscript has been judged scientifically suitable for publication and will be formally accepted for publication once it meets all outstanding technical requirements.

Kind regards,

Faten Abdelaal Okda

Academic Editor

PLOS ONE
---

## [Editor Report · Acceptance letter]

10 Jan 2023

PONE-D-22-17597R1 

Characterization of Two Linear Epitopes SARS CoV-2 Spike Protein Formulated in Tandem Repeat 

Dear Dr. Tarigan:

I'm pleased to inform you that your manuscript has been deemed suitable for publication in PLOS ONE. Congratulations! Your manuscript is now with our production department. 

Kind regards, 

on behalf of

Dr. Faten Abdelaal Okda 

Academic Editor

PLOS ONE